# Modality differences in ERP components between somatosensory and auditory Go/No-go paradigms in prepubescent children

Hiroki Nakata[1]*, Miho Takezawa[2], Keita Kamijo[3], Manabu Shibasaki[1]

**1** Department of Health Sciences, Faculty of Human Life and Environment, Nara Women's University, Nara, Japan, **2** The Elementary School Attached to Nara Women's University, Nara, Japan, **3** Faculty of Liberal Arts and Sciences, Chukyo University, Aichi, Japan

* hiroki-nakata@cc.nara-wu.ac.jp

**Data Availability Statement:** All relevant data are within the paper and its Supporting Information files.

## Abstract

We investigated modality differences in the N2 and P3 components of event-related potentials (ERPs) between somatosensory and auditory Go/No-go paradigms in eighteen healthy prepubescent children (mean age: 125.9±4.2 months). We also evaluated the relationship between behavioral responses (reaction time, reaction time variability, and omission and commission error rates) and amplitudes and latencies of N2 and P3 during somatosensory and auditory Go/No-go paradigms. The peak latency of No-go-N2 was significantly shorter than that of Go-N2 during somatosensory paradigms, but not during auditory paradigms. The peak amplitude of P3 was significantly larger during somatosensory than auditory paradigms, and the peak latency of P3 was significantly shorter during somatosensory than auditory paradigms. Correlations between behavioral responses and the P3 component were not found during somatosensory paradigms. On the other hand, in auditory paradigms, correlations were detected between the reaction time and peak amplitude of No-go-P3, and between the reaction time variability and peak latency of No-go-P3. A correlation was noted between commission error and the peak latency of No-go-N2 during somatosensory paradigms. Compared with previous adult studies using both somatosensory and auditory Go/No-go paradigms, the relationships between behavioral responses and ERP components would be weak in prepubescent children. Our data provide findings to advance understanding of the neural development of motor execution and inhibition processing, that is dependent on or independent of the stimulus modality.

## Introduction

Event-related potentials (ERPs) obtained by time-locked averaging electroencephalography (EEG) with high temporal resolution have been used to investigate the neural substrates of motor execution and inhibition during Go/No-go paradigms for over 30 years. Two

**Funding:** This study was supported by a Japan Society for the Promotion of Science KAKENHI Grant-in-Aid for Scientific Research C (19K11576) (to H. N.). The funders had no role in study design, data collection and analysis, decision to publish, or preparation of the manuscript.

**Competing interests:** No authors have competing interests.

components, a negative deflection at approximately 140–300 ms (N2 component) after stimulus onset and a positive deflection at approximately 300–600 ms (P3 component), elicited in No-go trials were larger than the ERPs recorded in Go trials in adults [1–3]. These No-go-related brain activities have mainly been investigated using visual and auditory stimuli, and only a few studies reported activity in the somatosensory (tactile) modality [4–6]. In somatosensory Go/No-go paradigms, the amplitude of the No-go-N2 component was also greater than that of the Go-N2 component, and the amplitude of No-go-P3 was larger than that of Go-P3. The enhanced No-go-related components, No-go-N2 and No-go-P3, reflect common neural activities specific to the inhibitory process, irrespective of the sensory modality.

Many studies also focused on the development of motor execution and inhibition processing in children [7–13]. As the characteristics of behavioral data during Go/No-go paradigms, the reaction time (RT) was longer in children than in adults, and error rates including omission (i.e., a slow response or no pushing with Go stimulus) and commission (i.e., error pushing with No-go stimulus) were higher in children than in adults [7, 12]. In ERP waveforms, the same sequence of components was generally elicited in both children and adults. The major difference in children was observed in a large frontal N2 overlaying the early components [10, 11]. Jonkman [9] using visual Go/No-go paradigms reported that No-go-N2 effects were largest and more widely distributed across fronto-parietal electrodes in children aged 6–7 years old, and that they decreased linearly with age. The second difference was an absence of the fronto-central No-go-P3. As mentioned above, in adults, the amplitude of No-go-P3 was generally larger than that of Go-P3, and No-go-P3 shows a more anterior distribution relative to Go-P3, the so-called 'anteriorization' of No-go-P3 [14–16]. However, this phenomenon was absent in children [10, 12]. These data suggest the immaturity of the fronto-parietal cortical-cortical network, and the immaturity for inhibition processing may cause a higher error rate, especially for commission errors [8].

In order to investigate modality differences of Go/No-go ERP waveforms in adults, previous studies compared the ERP waveforms in visual and auditory Go/No-go paradigms [14, 15, 17, 18]. For example, the amplitude of No-go-N2 was markedly smaller following auditory stimuli than after visual stimuli [14, 15, 19]. Falkenstein et al. [1] suggested that neural activity for inhibitory processing involved modality-specific differences, which was confirmed in a monkey study [20]. Recently, Yamashiro et al. [21] recorded ERPs during somatosensory and auditory Go/No-go paradigms from collegiate baseball players and track and field athletes. They showed significantly different ERP waveforms between the two groups during somatosensory Go/No-go paradigms, but not during auditory Go/No-go paradigms. They suggested that modality-specific neuroplastic changes took place with long-term skills training. After a thorough literature search, however, no study was found that examined the characteristics of somatosensory Go/No-go ERP waveforms in children, since the majority of previous studies used visual and auditory stimuli. Some previous studies using somatosensory-evoked potentials (SEPs) showed clear differences in waveforms [22] and the recovery function [23] between prepubescent children and adults, indicating an immature somatosensory system in children. In addition, based on the results of Yamashiro et al. [21], developmental differences between somatosensory and auditory cognitive processing might be observed among prepubescent children. To consider the developmental process of the somatosensory system, it is not enough to simply evaluate SEPs; it is also necessary to clarify the somatosensory processing in situations that require the prefrontal cortex function (i.e., cognitive task), which is a slow-developing brain region. Therefore, the main aim was to investigate modality differences between somatosensory and auditory Go/No-go paradigms among prepubescent children.

We also evaluated the relationship between the behavioral response and amplitudes and latencies of ERP components in somatosensory and auditory Go/No-go paradigms among

prepubescent children. The behavioral data, such as RT and error rates, and ERP components reflect different parameters in cognitive paradigms, but many previous studies reported close relationships. The latency of P3 has been considered a measure of the stimulus classification and evaluation speed [2, 24], and a correlation has been shown between RT and the peak latency of P3 [24–26]. Moreover, previous studies in adults reported correlations between RT and the amplitudes of No-go-N2 and No-go-P3 in visual and auditory Go/No-go paradigms [27–29], and between RT and the amplitude of No-go-P3 in a somatosensory Go/No-go paradigm [30, 31]. In addition, a higher commission error rate group would also be related to lower amplitudes of visual Go-P3 and No-go-P3, compared with a lower group, even though RTs do not differ between the two groups [32]. These data suggest that motor execution and inhibition in Go/No-go paradigms involves are closely related. By applying these findings, we hypothesized that the correlations between RT and the amplitudes of No-go-N2 and/or No-go-P3, and/or between error rates and the amplitudes of Go-P3 and/or No-go-P3 were observed in prepubescent children during both somatosensory and auditory Go/No-go paradigms. We also designed a Go (target) and No-go (non-target) stimulus with the same probability to avoid the effects of stimulus probability and minimize differences in response conflict between event types [4, 11–13, 33]. These data would advance understanding of the neural development of motor execution and inhibition processing, which is dependent on or independent of the stimulus modality.

## Materials and methods

### Participants

Eighteen normal prepubescent children (11 girls and 7 boys) with right handedness participated in this study. The mean age of the children in months was 125.9±4.2. No participants had a history of neurological or psychiatric disorders. Informed consent was obtained from all participants and their guardians. This study was approved by the Ethical Committee of Nara Women's University, Nara City, Japan.

### Task and procedure

The participants performed somatosensory and auditory Go/No-go paradigms. The order of conditions was randomized in each subject and counterbalanced across all participants. In the somatosensory Go/No-go paradigm, the Go stimulus was delivered to the second digit of the left hand, and the No-go stimulus to the fifth digit of the left hand with ring electrodes. The electrical stimulus used was a current constant square wave pulse of 0.2 ms in duration, and the stimulus intensity was 2 times the sensory threshold. In the auditory Go/No-go paradigm, auditory stimuli were presented binaurally through headphones (65-dB sound pressure level, 500-ms duration, 10-ms rise time, 10-ms fall time). Go and No-go stimuli were pure tones of 1,500 and 1,000 Hz, respectively.

Participants had to respond to the stimulus by pushing a button with their right thumb as quickly as possible only after presentation of the Go stimulus. The probability of Go and No-go stimuli was the same in a random series, with the interval of presentation being fixed at 2 sec. RT was measured for the Go stimulus. Each session comprised 120 epochs of stimulation, which included 60 epochs for the Go stimulus and 60 for the No-go stimulus. Participants kept their eyes open and focused on a small fixation point positioned in front of them at a distance of approximately 1 m throughout each task. As the error rate, omission and commission errors were separately calculated. In a practice run, participants were instructed to perform the Go/No-go paradigms for 20 stimuli before recording. In our previous study using the somatosensory Go/No-go paradigms, we set two conditions [4]. In one condition, the Go stimulus was

delivered to the second digit of the left hand, and the No-go stimulus to the fifth digit of the left hand. In the other condition, the Go and No-go stimuli were reversed in the left hand, i.e., Go and No-go stimuli were delivered to the fifth and second digits, respectively. As the results, no significant differences between conditions were observed in behavioral data including RT and error rates, nor in the peak amplitudes or latencies of somatosensory ERP components. Therefore, in the present study, we considered that the effects of physical differences between Go and No-go stimuli (i.e., stimulations to second and fifth digits in the left hand; pure tones of 1,500 and 1,000 Hz) on behavioral data and ERP components would be negligible.

## EEG recording

EEG was recorded with Ag/AgCl disk electrodes placed on the scalp at Fz, Cz, Pz, C3, and C4, according to the International 10–20 System. Each scalp electrode was referenced to linked earlobes, which were calculated as an averaged reference. In order to reject eye movements or blinks exceeding 100 μV, an electro-oculogram was recorded bipolarly with a pair of electrodes placed 2 cm lateral to the lateral canthus of the left eye and 2 cm above the upper edge of the left orbit and analyzed on-line. We also checked all raw data off-line, and if clear artifacts not exceeding 100 μV (ex., unexplained noise) were recorded, the trials were eliminated from averaging. Impedance was maintained at less than 5 kohm. All EEG signals were collected on a signal processor (Neuropack MEB-2300 system, Nihon-Kohden, Tokyo, Japan). The analysis epoch for ERPs was 800 ms, including a prestimulus baseline period of 100 ms. The bandpass filter was set at 0.1–50 Hz and the sampling rate was 1,000 Hz. In the somatosensory paradigms, the peak amplitudes and latencies of somatosensory N1 (N1s) and P3 components were measured at 110–230 and 260–600 ms, respectively. In the auditory paradigms, the peak amplitudes and latencies of auditory N1 (N1a) and P3 components were measured at 70–170 and 240–600 ms, respectively. In both Go/No-go paradigms, the peak amplitude and latency of the N2 component at Fz was measured at 200–330 ms. N1s and N1a components are recorded before N2, reflecting each sensory processing [4, 34–36]. Amplitudes were measured at baseline-to-peak. Slow responses exceeding 800 ms and incorrect responses were eliminated from averaging. In each paradigm, at least 30 trials or more were averaged. In total, 40.1±10.2 trials for somatosensory paradigms and 37.7±8.0 for auditory paradigms were averaged. As behavioral data, RT, the standard deviation (SD) of RT (i.e., reaction time variability), and omission (i.e., slow response or no pushing with Go stimulus) and commission errors (i.e., error pushing with No-go stimulus) were evaluated for each condition.

## Statistical analysis

As for behavioral data, RT, SD of RT, and omission and commission errors were compared between somatosensory and auditory paradigms using one-way repeated measures analysis of variance (ANOVA) with Modality (somatosensory vs. auditory) as a within-subject factor. Slow responses exceeding 800 ms were counted as omission errors.

The amplitudes and latencies of N1s and N1a components at C4 and Fz, respectively, were separately analyzed by one-way repeated measures ANOVA using the within-subject factor of Trial (Go vs. No-go), because N1s and N1a components were the largest at C4 and Fz among electrodes, respectively. The amplitudes and latencies of N2 components at Fz were separately submitted to repeated two-way measures ANOVA using the within-subject factors of Trial. Judging from grand-averaged ERP waveforms (Fig 1), the amplitude of N2 would be affected by overlapping of adjacent N1. Therefore, N2 components did not directly compare with Modality. The amplitudes and latencies of P3 components were separately submitted to repeated three-way measures ANOVA using the within-subject factors of Modality, Trial, and Electrode (middle; Fz, Cz, and Pz). In all repeated measures factors with more than two levels,

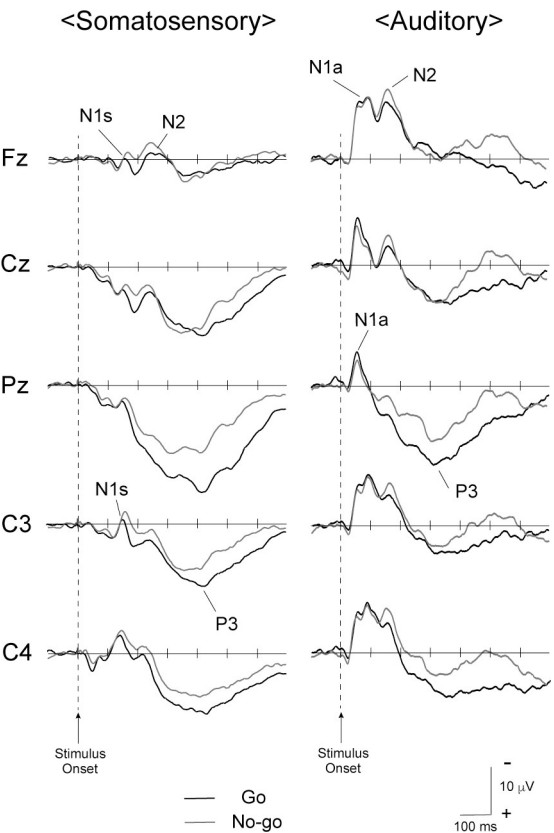

**Fig 1. Grand-averaged somatosensory and auditory ERP waveforms across all participants.** In the waveforms, the top is shown as negative, and the bottom is shown as positive.

we tested whether Mauchly's sphericity assumption was violated. If the result of Mauchly's test was significant and the assumption of sphericity was violated, Greenhouse-Geisser adjustment was used to correct the sphericity by altering the degrees of freedom using a correction coefficient epsilon. When significant effects were identified, Bonferroni post-hoc multiple-comparison was adjusted to identify specific differences.

We also analyzed the bivariate correlative relationship between behavioral responses and amplitudes and latencies of N2 and P3 at Fz, Cz, and Pz. This analysis was performed after checking data with a normal distribution using the Kolmogorov-Smirnov test. If a normal distribution was confirmed, Pearson's correlation was calculated. If non-parametric data were found, Spearman's correlation was analyzed. Considering Type I errors on correlation analysis with many data, we considered only significant data with an r value greater than ±0.500 [30]. Significance was set at $p < 0.05$.

# Results

## Behavioral data

Table 1 shows behavioral data on RT, SD of RT, and omission and commission errors. No significant effects were observed between somatosensory and auditory paradigms in any of behavioral measurements (RT: $F_{(1, 17)} = 0.735$, $p = 0.403$, $\eta^2 = 0.041$; SD of RT: $F_{(1, 17)} = 0.287$, $p = 0.604$, $\eta^2 = 0.016$; omission error: $F_{(1, 17)} = 1.471$, $p = 0.242$, $\eta^2 = 0.080$; commission error: $F_{(1, 17)} = 0.482$, $p = 0.497$, $\eta^2 = 0.028$).

**Table 1. Behavioral data during somatosensory and auditory Go/No-go paradigms with SE.**

|  | Somatosensory | Auditory |
|---|---|---|
| RT (ms) | 435 (24) | 415 (25) |
| SD of RT (ms) | 123 (8) | 120 (7) |
| Omission error (%) | 5.7 (1.3) | 4.1 (0.8) |
| Commission error (%) | 4.1 (0.6) | 3.5 (0.7) |

No significant differences were observed between modalities. RT = reaction time. SD = standard deviation.

## ERP components

Fig 1 shows grand-averaged waveforms of ERPs during somatosensory and auditory Go/No-go paradigms. The ERP components, N1s and N1a, were elicited during each paradigm. The mean values for amplitudes and latencies of N1s and N1a are listed in Table 2. In addition, N2 components were detected at Fz during Go and No-go stimuli during somatosensory and auditory paradigms. These components were confirmed in 14 of 18 children during the Go stimulus and 17 of 18 children during the No-go stimulus during somatosensory paradigms, and in all children for Go and No-go stimuli during auditory paradigms.

ANOVAs for the peak amplitude of N1s showed a significant main effect of Trial ($F_{(1, 15)}$ = 6.366, $p$ = 0.023, $\eta^2$ = 0.298), suggesting larger amplitude of No-go-N1s than that of Go-N1s. ANOVAs for the peak latency of N1s showed no significant main effect of Trial ($F_{(1, 15)}$ = 0.061, $p$ = 0.809, $\eta^2$ = 0.004). ANOVAs for the peak amplitude and latency of N1a showed no significant main effect of Trial, suggesting no differences between Go and No-go trials (N1a amplitude: $F_{(1, 14)}$ = 1.524, $p$ = 0.237, $\eta^2$ = 0.098; N1a latency: $F_{(1, 14)}$ = 0.042, $p$ = 0.840, $\eta^2$ = 0.098) (Table 2).

Regarding the peak amplitude of N2, no significant main effects of Trial were observed during somatosensory and auditory paradigms (somatosensory: $F_{(1, 12)}$ = 0.886, $p$ = 0.365, $\eta^2$ = 0.069; auditory: $F_{(1, 17)}$ = 0.929, $p$ = 0.349, $\eta^2$ = 0.052). ANOVA for the peak latency of N2 during somatosensory paradigms showed significant main effects of Trial ($F_{(1, 12)}$ = 5.401, $p$ = 0.038, $\eta^2$ = 0.310), suggesting a shorter latency of No-go-N2 than Go-N2, whereas no such difference was observed during auditory paradigms ($F_{(1, 17)}$ = 0.003, $p$ = 0.960, $\eta^2$ = 0.000) (Table 3).

The results of ANOVAs for the peak amplitude of P3 showed a significant main effect of Modality ($F_{(1, 17)}$ = 47.980, $p$ < 0.001, $\eta^2$ = 0.738), indicating a larger amplitude during somatosensory than auditory paradigms. ANOVAs also showed significant main effects of Trial ($F_{(1, 17)}$ = 5.087, $p$ = 0.038, $\eta^2$ = 0.230) and Electrode ($F_{(4, 68)}$ = 106.029, $p$ < 0.001, $\eta^2$ = 0.862), as well as Modality-Trial interaction ($F_{(1, 17)}$ = 6.533, $p$ = 0.020, $\eta^2$ = 0.278), Modality-Electrode interaction ($F_{(2, 34)}$ = 7.788, $p$ = 0.002, $\eta^2$ = 0.314), and Trial-Electrode

**Table 2. Average values for peak amplitudes and latencies of N1s and N1a components with SE.**

|  | Go | | | | | No-go | | | | |
|---|---|---|---|---|---|---|---|---|---|---|
|  | Fz | Cz | Pz | C3 | C4 | Fz | Cz | Pz | C3 | C4 |
| Somatosensory |  |  |  |  |  |  |  |  |  |  |
| N1s amplitude (μV) | -3.0 (0.9) | 0.3 (1.5) | 2.7 (1.4) | -2.6 (1.0) | -3.8 (0.7) | -3.5 (1.2) | -1.1 (1.0) | -1.2 (1.1) | -4.8 (0.9) | -6.4 (0.9) |
| N1s latency (ms) | 167 (5) | 167 (6) | 161 (5) | 160 (5) | 154 (6) | 168 (5) | 172 (6) | 165 (3) | 170 (5) | 155 (5) |
| Auditory |  |  |  |  |  |  |  |  |  |  |
| N1a amplitude (μV) | -13.5 (1.5) | -10.7 (1.2) | -7.2 (0.8) | -11.9 (1.5) | -11.8 (1.3) | -12.6 (1.4) | -9.8 (1.4) | -6.6 (1.3) | -11.7 (1.5) | -11.6 (1.3) |
| N1a latency (ms) | 121 (6) | 109 (6) | 101 (6) | 117 (7) | 118 (7) | 121 (6) | 110 (6) | 102 (6) | 121 (6) | 126 (5) |

**Table 3. Average values for N2 and P3 in Go and No-go stimulus during somatosensory and auditory Go/No-go paradigms with SE.**

|  | Go | | | | | No-go | | | | |
|---|---|---|---|---|---|---|---|---|---|---|
|  | Fz | Cz | Pz | C3 | C4 | Fz | Cz | Pz | C3 | C4 |
| N2 amplitude (μV) | | | | | | | | | | |
| Somatosensory | -3.4 (1.1) | | | | | -4.8 (1.0) | | | | |
| Auditory | -12.4 (1.6) | | | | | -13.6 (0.9) | | | | |
| N2 latency (ms) | | | | | | | | | | |
| Somatosensory | 266 (6) | | | | | 253 (6) | | | | |
| Auditory | 241 (8) | | | | | 241 (8) | | | | |
| P3 amplitude (μV) | | | | | | | | | | |
| Somatosensory | 7.4 (1.1) | 21.1 (1.6) | 28.5 (1.7) | 16.5 (1.3) | 17.9 (1.5) | 8.5 (1.2) | 17.6 (1.5) | 18.2 (1.7) | 11.7 (1.1) | 12.0 (1.3) |
| Auditory | 2.6 (1.3) | 10.5 (1.3) | 16.9 (1.2) | 8.3 (1.8) | 10.5 (1.4) | 4.9 (0.8) | 11.4 (1.3) | 13.0 (1.3) | 6.6 (1.0) | 7.3 (1.2) |
| P3 latency (ms) | | | | | | | | | | |
| Somatosensory | 393 (16) | 394 (17) | 377 (16) | 400 (20) | 400 (20) | 378 (18) | 350 (14) | 354 (21) | 369 (16) | 393 (19) |
| Auditory | 446 (24) | 442 (18) | 402 (17) | 421 (15) | 473 (23) | 442 (229) | 416 (17) | 437 (15) | 434 (20) | 440 (19) |

interaction (F (2, 34) = 30.570, p < 0.001, $\eta^2$ = 0.643). A post-hoc Bonferroni test of the Trial-Electrode interaction showed that the peak amplitudes of Go-P3 were significantly larger at Pz than at Fz and Cz (p < 0.001, respectively), and at Cz than Fz (p < 0.001, respectively). In contrast, the peak amplitudes of No-go-P3 during auditory and somatosensory paradigms were significantly larger at Pz than at Ft and Cz (p < 0.001, respectively). These data indicate that No-go-P3 has a more anterior distribution relative to Go-P3, the so-called 'anteriorization' of No-go-P3. In addition, we examined the normalized amplitude values to clarify the differences in the scalp topography [37]. ANOVAs revealed a significant main effect of Electrode (F (2, 34) = 178.071, p < 0.001, $\eta^2$ = 0.913), and Trial-Electrode interaction (F (2, 34) = 6.320, p = 0.005, $\eta^2$ = 0.271). This interaction indicates the different distribution between Go and No-go trials (Fig 2). However, the three two-way interactions were superseded by a three-way interaction of Modality-Trial-Electrode (F (2, 34) = 4.498, p = 0.018, $\eta^2$ = 0.209). Decomposition of the three-way interaction facilitated examination of Modality-Stimulus within each electrode and revealed a Modality-Stimulus interaction only at Pz (F (1, 17) = 11.193, p = 0.04, $\eta^2$ = 0.397). This two-way interaction indicated that the peak amplitude of Go-P3 was larger than that of No-go-P3 during the somatosensory paradigms (F (1, 17) = 15.310, p = 0.001, $\eta^2$ = 0.474), whereas no such difference was observed during the auditory paradigms (F (1, 17) = 1.003, p = 0.331, $\eta^2$ = 0.056).

ANOVAs for the peak latency of P3 showed a significant main effect of Modality (F (1, 17) = 23.925, p < 0.001, $\eta^2$ = 0.585) with a shorter peak latency during somatosensory (mean = 374 ms, SD = 42) than auditory (mean = 431 ms, SD = 50) paradigms. No other main effects nor interactions were observed (Table 3).

### Relationship between RT and N2 and P3

In auditory Go/No-go paradigms, a significant negative correlation was noted between RT and the peak amplitude of No-go-P3 at Pz (r = -0.545, p = 0.019), indicating that, for children with a shorter RT, the peak amplitude of No-go-P3 was larger. Except for this, no other correlations were observed between RT and N2 and P3 (Table 4).

### Relationship between SD of RT and N2 and P3

In auditory Go/No-go paradigms, significant positive correlations were observed between SD of RT and the peak latency of No-go-P3 at Fz (r = 0.610, p = 0.007) and at Cz (r = 0.597,

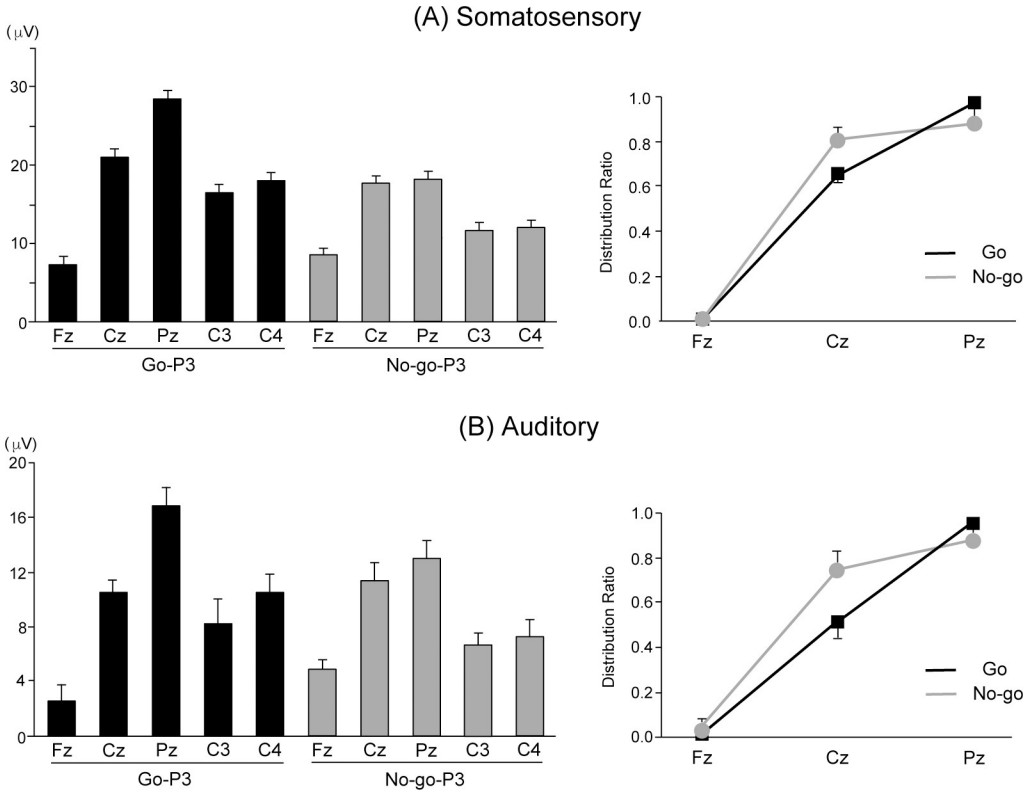

**Fig 2.** Mean values for the amplitudes of Go-P3 and No-go-P3 and the distribution ratio at the midline electrodes in (A) somatosensory and (B) auditory Go/No-go paradigms. Vertical lines indicate standard error (SE).

p = 0.009), indicating that, for children with a smaller SD of RT, the peak latency of No-go-P3 was shorter. Except for these, no other correlations were observed between SD of RT and N2 and P3 (Table 4).

### Relationship between omission error and N2 and P3

No correlations were observed between omission error and N2 and P3 (Table 4).

### Relationship between commission error and N2 and P3

In somatosensory Go/No-go paradigms, a significant positive correlation was observed between commission error and the peak latency of No-go-N2 (r = 0.645, p = 0.005), indicating that, for children with a lower commission error rate, the peak latency of No-go-N2 was shorter. Except for this, no other correlations were observed between RT and N2 and P3 (Table 4).

## Discussion

The present study evaluated modality differences in ERP components between somatosensory and auditory Go/No-go paradigms among prepubescent children. We also evaluated the relationship between behavioral responses (RT, SD of RT, and omission and commission error rates) and the amplitudes and latencies of N2 and P3 during somatosensory and auditory Go/No-go paradigms.

**Table 4. Correlation matrix between behavioral data and amplitude and latency of ERP components.**

| | Go | | | No-go | | |
|---|---|---|---|---|---|---|
| | Fz | Cz | Pz | Fz | Cz | Pz |
| RT | | | | | | |
| Amplitude | | | | | | |
| Somatosensory-N2 | 0.279 | | | 0.096 | | |
| Auditory-N2 | -0.164 | | | -0.130 | | |
| Somatosensory-P3 | -0.142 | -0.246 | -0.079 | 0.040 | 0.174 | 0.080 |
| Auditory-P3 | 0.103 | -0.302 | -0.452 | -0.248 | -0.319 | -0.545 * |
| Latency | | | | | | |
| Somatosensory-N2 | -0.248 | | | -0.285 | | |
| Auditory-N2 | -0.362 | | | 0.203 | | |
| Somatosensory-P3 | 0.095 | 0.272 | 0.121 | -0.034 | -0.111 | -0.011 |
| Auditory-P3 | -0.247 | -0.269 | 0.014 | 0.267 | 0.189 | -0.008 |
| SD of RT | | | | | | |
| Amplitude | | | | | | |
| Somatosensory-N2 | 0.104 | | | -0.018 | | |
| Auditory-N2 | -0.399 | | | -0.226 | | |
| Somatosensory-P3 | -0.225 | -0.358 | -0.303 | 0.198 | -0.059 | -0.166 |
| Auditory-P3 | 0.06 | -0.347 | -0.480 | 0.332 | -0.407 | -0.427 |
| Latency | | | | | | |
| Somatosensory-N2 | 0.154 | | | -0.032 | | |
| Auditory-N2 | -0.325 | | | 0.395 | | |
| Somatosensory-P3 | 0.265 | 0.304 | -0.006 | 0.254 | 0.051 | 0.152 |
| Auditory-P3 | -0.016 | 0.244 | 0.399 | 0.610 ** | 0.597 ** | 0.187 |
| Omission error | | | | | | |
| Amplitude | | | | | | |
| Somatosensory-N2 | -0.311 | | | 0.036 | | |
| Auditory-N2 | -0.490 | | | -0.424 | | |
| Somatosensory-P3 | -0.470 | -0.442 | -0.242 | -0.178 | -0.271 | -0.211 |
| Auditory-P3 | -0.068 | -0.29 | -0.066 | 0.291 | 0.051 | -0.057 |
| Latency | | | | | | |
| Somatosensory-N2 | 0.051 | | | 0.246 | | |
| Auditory-N2 | 0.214 | | | -0.326 | | |
| Somatosensory-P3 | -0.081 | 0.284 | 0.355 | 0.292 | -0.139 | 0.032 |
| Auditory-P3 | 0.019 | 0.048 | 0.115 | 0.368 | 0.339 | 0.089 |
| Commission error | | | | | | |
| Amplitude | | | | | | |
| Somatosensory-N2 | -0.355 | | | -0.432 | | |
| Auditory-N2 | -0.108 | | | -0.254 | | |
| Somatosensory-P3 | 0.021 | 0.095 | -0.033 | 0.042 | 0.055 | -0.141 |
| Auditory-P3 | 0.05 | -0.3 | 0.201 | 0.368 | -0.006 | 0.354 |
| Latency | | | | | | |
| Somatosensory-N2 | 0.409 | | | 0.645 ** | | |
| Auditory-N2 | 0.008 | | | -0.226 | | |
| Somatosensory-P3 | 0.342 | 0.172 | -0.258 | -0.239 | 0.097 | -0.01 |
| Auditory-P3 | 0.416 | 0.135 | 0.015 | -0.039 | -0.201 | -0.050 |

* $p < 0.05$; *: $p < 0.01$.

## Behavioral data

RT is an important measure for understanding sensorimotor performance in humans [38], and is defined as the time from stimulus onset to the response, including components such as stimulus evaluation and response selection [39]. The SD of RT is often used to evaluate the reaction time variability of the time from stimulus onset to the response [40, 41]. As a cognitive model, it is known that omission error is associated with increased attention to punishment, and commission error is associated with increased attention to reward [42]. In our previous study, in which we used the same experimental setting as this study with young adults [43], during somatosensory and auditory Go/No-go paradigms, the mean RTs were 307±50 and 311±52 ms, respectively, and SDs of RT were 75±16 and 77±23 ms, respectively. The percentages of omission and commission errors in the somatosensory Go/No-go paradigms were 1.1 ±1.3 and 1.0±0.9%, respectively, and those in the auditory Go/No-go paradigms were 1.9±2.3 and 1.0±0.6%, respectively [43]. These data suggest no significant difference in behavioral responses between somatosensory and auditory Go/No-go paradigms using young adults. Consistent with this, the present study showed no significant differences in behavioral responses between somatosensory and auditory Go/No-go paradigms (Table 1). On the other hand, RT was longer in prepubescent children than in adults during somatosensory and auditory paradigms, SD of RT was larger in prepubescent children than in adults, and rates of omission and omission errors were higher in prepubescent children than in adults. These results were consistent with other previous studies [8, 9], and suggest an immature cognitive function reflected by behavioral responses in prepubescent children, even if they performed somatosensory Go/No-go paradigms.

## N1 component

The amplitude and latency of N1a did not differ between Go and No-go stimuli in auditory Go/No-go paradigms (Table 2), being consistent with previous studies in children [11, 13]. As reported in the literature, N1a, commonly elicited by simple auditory stimuli, is a composite of multiple components [44] with generators around Heschl's gyrus and the planum temporale [35, 36], and the frontal cortex [45, 46]. In the time range recoded in N1a for adults, a previous study reported different topographic distributions between Go and No-go trials [47], suggesting the involvement of different cortical processing. In prepubescent children, a similar N1a response to both Go and No-go stimuli may be compatible with similar levels of sensory processing.

The amplitude of N1s at C4 was significantly larger in No-go trials than in Go trials in somatosensory Go/No-go paradigms (Table 2). Previous studies using somatosensory Go/No-go paradigms in adults also showed that the amplitude of No-go-N1s was larger than that of Go-N1s, and that the latency of No-go-N1s was later than that of Go-N1s [4, 16, 30]. Previous studies reported that N1s, which is often called N140, was generated from several regions including the secondary somatosensory cortex, insula, cingulate cortex, and prefrontal cortex [4, 34, 48, 49]. Moreover, Nakata et al. [4] using magnetoencephalography (MEG) indicated that No-go-N1s involved these generator mechanisms for N1s as well as No-go-specific neural activity from the prefrontal cortex (PFC). In other words, when adults performed somatosensory Go/No-go paradigms, the overlapped potentials led to a larger amplitude of No-go-N1s than that of Go-N1s. On the other hand, as shown in Fig 1, N1s and N2 components in prepubescent children were separated. Thus, the characteristics of N2 in somatosensory Go/No-go paradigms are separately discussed below.

## N2 components

The functional significance of N2 has been a matter of debate in adults' data during Go/No-go paradigms. N2 is clearly recorded during visual Go/No-go paradigms rather than auditory Go/No-go paradigms [14, 15, 18]. The neural activities of N2 may include the processing of motor inhibition at PFC [27, 49] as well as conflict monitoring at the anterior cingulate cortex (ACC) [50, 51]. As for the characteristics of N2 in children, Jonkman [9] showed that the amplitude of N2 diminished gradually from young childhood through to adulthood.

This is the first study to detect the frontal N2 among prepubescent children, even though a somatosensory Go/No-go paradigm was used. Indeed, N1s and N2 components in prepubescent children were separated (Fig 1), but we considered that N1s and frontal N2 components in a somatosensory Go/No-go paradigm gradually overlapped with increasing age. In data from adults, the amplitude of N1s was the largest at Fz (i.e., frontal electrode) rather than at C4 (i.e., lateral electrode) [52]. The difference of distribution in N1s between prepubescent children and adults may reflect the developmental process for motor execution and inhibition. To clarify this, further studies including age-related changes are needed. Furthermore, we found that the peak latency during somatosensory paradigms was significantly earlier in No-go-N2 than in Go-N2, but was not during auditory paradigms (Table 3). In a study of monkeys, Gemba and Sasaki [20] observed No-go-related neural activities after an auditory stimulus in the rostral part of the dorsal bank of the principal sulcus, as opposed to the caudal part of the same bank after a visual stimulus. The present study did not directly address the differences in generator mechanisms of frontal N2 between somatosensory and auditory paradigms, but our findings suggest that the appearance itself of frontal N2 among children does not depend on sensory modalities, and the strength and speed of neural activities involve modality differences. In addition, the difference in the peak latency of No-go-N2 might be related to the developmental difference between somatosensory and auditory processing of motor inhibition.

## P3 components

We showed that the amplitudes of P3 were significantly larger during somatosensory than auditory Go/No-go paradigms (Table 3), suggesting that neural activities for motor execution and inhibition were larger during somatosensory than auditory Go/No-go paradigms. Regarding adults' data, Imanaka et al. [43] reported that the amplitudes of P3 were significantly larger during somatosensory than auditory Go/No-go paradigms, which was consistent with our findings. Falkenstein et al. [15] also showed that amplitudes of P3 were significantly larger during visual than auditory Go/No-go paradigms. Studies of adults suggested that neural activities and strengths in motor execution and inhibition were smaller in auditory paradigms than in visual and somatosensory paradigms. Our data indicate the existence of a modality difference in the amplitude of P3 even in prepubescent children. In other words, although it is unclear whether this phenomenon is innate or a developmental process, modality differences in P3 between somatosensory and auditory Go/No-go paradigms are already present in prepubescent children.

We also showed that the peak amplitudes of P3 were mainly distributed at centro-parietal electrodes in prepubescent children compared with adults, even though the weak 'anteriorization' of No-go-P3 was observed in prepubescent children. Some previous studies already suggested an underlying mechanism whereby in young children, functional proficiency in the detection of Go stimuli is achieved earlier than in inhibition of responses to No-go stimuli [8]. Logan & Cowan [53] proposed that the mechanisms governing inhibition and attention to targets may function independently to some degree. The larger amplitude of Go-P3 in prepubescent children may be related to the developmental process in neural activity for motor execution and inhibition.

The latency of P3 was significantly shorter during somatosensory than auditory Go/No-go paradigms (Table 3), suggesting that the stimulus classification and evaluation speed were shorter during somatosensory than auditory Go/No-go paradigms. These data were inconsistent with previous findings from adults. Imanaka et al. [43] showed no significant differences in latencies of P3 between the somatosensory and auditory Go/No-go paradigms. Falkenstein et al. [15] showed that latencies of P3 were significantly shorter during auditory than visual Go/No-go paradigms. We inferred that the latencies of N2 and P3 and mean RT were influenced easily by some factors, such as the subject and stimulus conditions. For example, Falkenstein et al. [14, 15] reported contradictory findings even when they used similar experimental paradigms. Nieuwenhuis et al. [17] also suggested that RT was associated with context letters. In their paradigms, when the context letter looked similar but sounded different (i.e., Go trial = T, No-go trial = F), RTs were shorter during the auditory Go/No-go paradigm than visual Go/No-go paradigm. In contrast, when the context letter looked different but sounded similar (i.e., Go trial = S, No-go trial = F), RTs were faster in the visual Go/No-go paradigm than auditory Go/No-go paradigm. These factors also might be applied for prepubescent children.

In addition, when comparing the present data with data from our previous adult study on the peak latency of P3 [43], the peak latencies of P3 during somatosensory and auditory Go/No-go paradigms were clearly longer in prepubescent children than adults. These suggest the immaturity of stimulus classification and evaluation speed in prepubescent children.

## Relationship between behavioral responses and ERP components

The amplitude is smaller during the auditory Go/No-go paradigm than the somatosensory Go/No-go paradigm, but it currently remains unclear whether neural functions relating to motor execution and inhibition processing differ between somatosensory and auditory Go/No-go paradigms. Based upon the present results (Fig 1 and Tables 3 and 4), we propose that the smaller amplitude of P3 in the auditory Go/No-go paradigm more sensitively reflects the neural functions of motor execution and inhibition. Moreover, these results may indicate that relationships between behavioral responses and ERP components might be stronger in No-go than Go trials, even though behavioral responses such as RT and SD of RT were recorded from Go trials. Similar results were observed in previous studies on adults among somatosensory [30] and visual and auditory Go/No-go paradigms [18]. Given these results, coupled with our previous studies in adults using functional magnetic resonance imaging (fMRI) and MEG during somatosensory Go/No-go paradigms [49, 54], the greater activation of the dorsolateral (DLPFC) and ventrolateral prefrontal cortices (VLPFC), anterior cingulate cortex (ACC), inferior parietal lobule, and caudate due to No-go relative to Go trials may be correlated with the results of behavioral responses. That is, behavioral responses might be closely linked to the strength of neural activity for motor inhibition rather than motor execution.

In somatosensory Go/No-go paradigms, a correlation was only found between commission error and the peak latency of No-go-N2. Behavioral responses including RT, SD of RT, and error rates, and the peak amplitude and latency of ERPs indicate different indices in human information processing, but many previous studies in adults reported correlations between behavioral responses and ERP components during visual, auditory, and somatosensory Go/No-go paradigms [27–32]. In other words, the construction of relationships between actual behavioral responses and neural activities reflected by ERPs in adults may reflect the maturity of the cognitive function. Taking these previous studies into consideration, such a relationship between them would be weaker in prepubescent children than in adults, suggesting an immaturity of the cognitive function in prepubescent children.

### Limitation of the present study

The present study used peak measurements for each ERP component. However, since peak measurements may involve high-frequency noises (ex., double peaks of P3), the definition to determine the peak amplitude and latency would be important in this study. In addition, Fig 1 shows simply grand-averaged ERP waveforms across all participants. Therefore, for example, if there are subjects with a large and shorter P3 amplitude, the ERP waveforms could be distorted. This may be related to the difference in appearance between the actual value (i.e., tables) and waveforms (i.e., figures). Finally, we did not directly compare the data between prepubescent children and adults. Thus, further studies are needed to clarify the detailed differences in neural mechanisms between them.

## Conclusion

This is the first study to examine modality differences in ERP waveforms between somatosensory and auditory Go/No-go paradigms in prepubescent children. The frontal N2 component was specifically recorded among prepubescent children using visual and auditory Go/No-go paradigms [7–13], and our data showed that this component was detected even with a somatosensory Go/No-go paradigm. This finding suggests that the existence of the frontal N2 does not depend on sensory modalities. In addition, since N1s and N2 components were separately recorded (Fig 1) and the characteristics of N1s among prepubescent children differed from those of adults, we propose that the changes in the frontal negative potentials with increasing age reflect the developmental process for motor execution and inhibition. In a direct comparison of P3 between somatosensory and auditory Go/No-go paradigms, the amplitudes of P3 were significantly larger during somatosensory than auditory Go/No-go paradigms (Table 3). It is unclear whether this phenomenon is innate or a developmental process, but our data suggest that modality differences in P3 between somatosensory and auditory Go/No-go paradigms were already present in prepubescent children.

We also evaluated the relationship between behavioral responses (i.e., RT, SD of RT, and omission and commission errors) and N2 and P3 components. Based upon the present results (Tables 3 and 4), the characteristics of P3 components in prepubescent children included modality differences between somatosensory and auditory Go/No-go paradigms, and we propose that the smaller amplitude of P3 in the auditory Go/No-go paradigm more sensitively reflects the neural functions of motor execution and inhibition. Our data provide findings to advance understanding of the neural development of motor execution and inhibition processing, which is dependent on or independent of the stimulus modality.

## Supporting information

**S1 Data. Individual data.**
(XLSX)

## Author Contributions

**Conceptualization:** Hiroki Nakata, Miho Takezawa, Keita Kamijo, Manabu Shibasaki.

**Data curation:** Hiroki Nakata, Miho Takezawa.

**Formal analysis:** Hiroki Nakata, Keita Kamijo.

**Funding acquisition:** Hiroki Nakata, Keita Kamijo.

**Investigation:** Hiroki Nakata, Miho Takezawa.

**Methodology:** Hiroki Nakata, Manabu Shibasaki.

**Project administration:** Hiroki Nakata.

**Resources:** Hiroki Nakata.

**Supervision:** Keita Kamijo, Manabu Shibasaki.

**Validation:** Keita Kamijo.

**Visualization:** Hiroki Nakata.

**Writing – original draft:** Hiroki Nakata, Keita Kamijo, Manabu Shibasaki.

**Writing – review & editing:** Hiroki Nakata, Keita Kamijo, Manabu Shibasaki.

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
