## [Decision Letter · Decision Letter 0]

4 Aug 2021

PONE-D-21-09252

Modality differences in ERP components between somatosensory and auditory Go/No-go paradigms in prepubescent children

PLOS ONE

Dear Dr. Nakata,

Thank you for submitting your manuscript to PLOS ONE. After careful consideration, we feel that it has merit but does not fully meet PLOS ONE’s publication criteria as it currently stands. Therefore, we invite you to submit a revised version of the manuscript that addresses the points raised during the review process.

This is an interesting paper. However, as the authors can see both reviewers ha raised major and minor issues that must be addressed. Thus, I suggest the authors revise their manuscript according to reviewers' comments and submit it for further evaluation.

We look forward to receiving your revised manuscript.

Kind regards,

Vilfredo De Pascalis

Academic Editor

PLOS ONE

Additional Editor Comments (if provided):

This is an interesting paper. However, as the authors can see both reviewers ha raised major and minor issues that must be addressed. Thus, I suggest the authors revise their manuscript according to reviewers' comments and submit it for further evaluation.

Journal Requirements:

2. Please provide additional details regarding participant consent. In the Methods section, please ensure that you have specified (1) whether consent was informed and (2) what type you obtained (for instance, written or verbal).

This study was supported by a Japan Society for the Promotion of Science KAKENHI Grant-in-Aid for Scientific Research C (19K11576) (to H. N.).

Reviewers' comments:

Reviewer's Responses to Questions

**Comments to the Author**

1. Is the manuscript technically sound, and do the data support the conclusions?

Reviewer #1: Partly

Reviewer #2: Partly

2. Has the statistical analysis been performed appropriately and rigorously? 

Reviewer #1: Yes

Reviewer #2: No

3. Have the authors made all data underlying the findings in their manuscript fully available?

Reviewer #1: Yes

Reviewer #2: No

4. Is the manuscript presented in an intelligible fashion and written in standard English?

Reviewer #1: Yes

Reviewer #2: Yes

5. Review Comments to the Author

Reviewer #1: Thank you for gave my the possibility to read this interesting paper. The manuscript was aimed to investigate the modality differences of Go/No-go ERP waveforms adults in prepubescent children vs. adults; secondarly the relationship between the behavioral response and amplitudes and latencies of ERP components in somatosensory and auditory Go/No-go paradigms among prepubescent children.

The manuscript is well wrote and easy to read however I have important questions to report:

My first concern is about the main hypothesis: in order to satisfy it they could compare a sample of prepubescent children vs. a sample of adult with the same procedure. It is not appropriate to test an hypothesis only comparing own data with other samples in previously published.

I suggest the Authors to report only the second hypothesis on the relationship between the behavioral response and amplitudes and latencies of ERP components in somatosensory and auditory Go/No-go paradigms among prepubescent children and discuss the results comparing with adults samples from literature in the discussion section. Finally they may report the lack of an adult sample in the limitation section.

My minor questions are the following:

The hypothesis is too large. It could be syntetized.

In the method section the Authors could report more information about the EEG system; how many electrodes; how many trials for each condition were considered for the analyses.

In the results section the authors could report the real p value also for non significant effects, and the effect size for the significant effects.

In the discussion section the authors could majorly articulate the interpretation about the immaturity of the cognitive function in prepubescent children.

Reviewer #2: Comments to the Authors:

The purpose of the present study was to investigate modality differences in the N2 and P3 event-related potentials (ERPs) between somatosensory and auditory Go/No-go paradigms in prepubescent children (n = 18). The results generally support the immaturity of motor execution and inhibition processing in prepubescent children.

Major:

As far as I read the manuscript, I have an impression that the authors would want to systematically report the N1, N2, and P3 results in somatosensory and auditory Go/No-go paradigms with the same stimuli and analyses as those in their previous study. Although I understand such situation, most readers would feel that (1) stimuli used in the present study are not well verified, (2) data analysis applied in this study are out-of-date, (3) the interpretation of the data is rather superficial and the conclusion (i.e., immaturity of motor execution and inhibition processing in prepubescent children) is less novel, and (4) new theoretical implication is not very clear, regardless of the fairly large volume of reported data. This paper would need modifications in these respects.

When analyzing and interpreting the data, the authors could be careful about at least for the following three points. First, the authors could be more careful about the overlapping of adjacent ERP components. For example, it was reported that somatosensory N2 at Fz was smaller than auditory N2 at Fz. However, as can be seen in Figure 1, the auditory N2 was immediately preceded by N1 that had the maximum amplitude at the same electrode (i.e., Fz); in contrast, somatosensory N2 was also preceded by N1 that had the maximum amplitude at a different electrode (i.e., C4). Given such a difference, a fair comparison between the somatosensory and auditory N2 seems to be difficult. Second, the authors should carefully discuss how the physical difference of the Go stimulus and No-go stimulus can affect the ERP and behavioral results. For example, readers would simply wonder how physical differences between the Go stimulus and No-go stimulus can be equalize between the auditory (i.e., tones of 1000 and 1500 Hz) and somatosensory modalities (i.e., stimulations to second and fifth digits in the left hand) and how such physical differences can affect ERP and behavioral results. This point should be carefully discussed and properly operationalized in the Introduction. Third, I would not prefer the use of “peak” measures, since it is well known that peak measures are highly vulnerable to high-frequency noises. Although “peak latency” measures are difficult to be replaced with other alternative measures, at least, the authors may use “mean amplitude” measures rather than “peak amplitude”. Anyway, if the authors use “peak” measures, then they should be careful about adverse influences due to high-frequency noises. For example, it was reported that the peak latency of somatosensory P3 was shorter than that of auditory P3 (lines 316-320). This is not consistent with the waveforms in Figure 1. I would suspect that this might be adverse influences of the use of peak latency measure.

Minor:

Results section (somatosensory N1): As can be seen in Figure 1, somatosensory N1 has the maximum amplitude at the right central electrode (i.e., C4). This seems natural, given that the somatosensory stimuli were presented to the participants’ fingers of the left hand. However, the size and timing of somatosensory N1 were quantified for the Fz electrode. This seems unreasonable.

Results section (amplitude of P3): It was reported that the peak amplitude of No-go P3 was larger than that of Go P3 during the somatosensory paradigm (lines 312-314). This is not consistent with the waveforms in Figure 1 as well as the values listed in Table 3. Is it correct?

Results section (scalp distributions): When examining between-condition differences of scalp topographies (e.g., lines 305-307 and 345-348), it is recommended to normalize the amplitude values, for example, by using the method of McCarthy & Wood (1985, Electroencephalography and Clinical Neurophysiology).

Results section (general): “p > .05” (where statistical differences were not significant) is not informative for readers. It would be better to simply show p-values. Also, effect sizes should be shown for all statistical tests.

Figure 1: ERPs are shown in a “negative-up” manner. This should be explicitly described in the figure or the figure caption.

Figure 1: Please show the baseline (i.e., x-axis) with time scales in each of ERP waveforms. It should be helpful for readers to visually evaluate the size and timing of each ERP component.

Limitation of the present study: I think that the lack of visual Go/No-go paradigm would not be a limitation of the present study.

6. PLOS authors have the option to publish the peer review history of their article (what does this mean?). If published, this will include your full peer review and any attached files.

Reviewer #1: **Yes: **Carlo Lai

Reviewer #2: No

---

## [Author Response · Author response to Decision Letter 0]

7 Sep 2021

Response to Reviewer #1

Thank you for gave my the possibility to read this interesting paper. The manuscript was aimed to investigate the modality differences of Go/No-go ERP waveforms adults in prepubescent children vs. adults; secondarly the relationship between the behavioral response and amplitudes and latencies of ERP components in somatosensory and auditory Go/No-go paradigms among prepubescent children. The manuscript is well wrote and easy to read however I have important questions to report:

My first concern is about the main hypothesis: in order to satisfy it they could compare a sample of prepubescent children vs. a sample of adult with the same procedure. It is not appropriate to test an hypothesis only comparing own data with other samples in previously published. I suggest the Authors to report only the second hypothesis on the relationship between the behavioral response and amplitudes and latencies of ERP components in somatosensory and auditory Go/No-go paradigms among prepubescent children and discuss the results comparing with adults samples from literature in the discussion section.

Thank you for this suggestion. Based on that, the first hypothesis was excluded, and the related paragraphs were revised.

Finally they may report the lack of an adult sample in the limitation section.

We added this as a limitation of the present study (page 24, lines 575-578):

“Finally, we did not directly compare the data between prepubescent children and adults. Thus, further studies are needed to clarify the detailed differences in neural mechanisms between them.”

My minor questions are the following: The hypothesis is too large. It could be syntetized.

We revised the Introduction section (from page 5, line 105 to page 6, line 147).

In the method section the Authors could report more information about the EEG system; how many electrodes; how many trials for each condition were considered for the analyses. 

Added (page 9, lines 203-206):

“EEG was recorded with Ag/AgCl disk electrodes placed on the scalp at Fz, Cz, Pz, C3, and C4, according to the International 10-20 System. Each scalp electrode was referenced to linked earlobes, which were calculated as an averaged reference.”

(page 10, lines 226-229):

“In each paradigm, at least 30 trials or more were averaged. In total, 40.1±10.2 trials for somatosensory paradigms and 37.7±8.0 for auditory paradigms were averaged.”

In the results section the authors could report the real p value also for non significant effects, and the effect size for the significant effects.

We revised the Result section (from page 12, line 274 to page 16, line 370).

In the discussion section, the authors could majorly articulate the interpretation about the immaturity of the cognitive function in prepubescent children.

Reviewer #2 also pointed out the similar problem. We revised the many parts of Discussion section.

Response to Reviewer #2

The purpose of the present study was to investigate modality differences in the N2 and P3 event-related potentials (ERPs) between somatosensory and auditory Go/No-go paradigms in prepubescent children (n = 18). The results generally support the immaturity of motor execution and inhibition processing in prepubescent children.

Major:

As far as I read the manuscript, I have an impression that the authors would want to systematically report the N1, N2, and P3 results in somatosensory and auditory Go/No-go paradigms with the same stimuli and analyses as those in their previous study. Although I understand such situation, most readers would feel that (1) stimuli used in the present study are not well verified, (2) data analysis applied in this study are out-of-date, (3) the interpretation of the data is rather superficial and the conclusion (i.e., immaturity of motor execution and inhibition processing in prepubescent children) is less novel, and (4) new theoretical implication is not very clear, regardless of the fairly large volume of reported data. This paper would need modifications in these respects.

Thank you for the constructive comment, and we revised many parts of manuscript. (1) We revised the Introduction section to understand the modality differences between somatosensory and auditory Go/No-go paradigms. (2) Reviewer #1 also pointed out the same issue. We revised the retrospective analysis with adults’ data. (3) & (4) We revised the Introduction and Discussion sections to show our novelty, hypothesis, and significance.

When analyzing and interpreting the data, the authors could be careful about at least for the following three points. First, the authors could be more careful about the overlapping of adjacent ERP components. For example, it was reported that somatosensory N2 at Fz was smaller than auditory N2 at Fz. However, as can be seen in Figure 1, the auditory N2 was immediately preceded by N1 that had the maximum amplitude at the same electrode (i.e., Fz); in contrast, somatosensory N2 was also preceded by N1 that had the maximum amplitude at a different electrode (i.e., C4). Given such a difference, a fair comparison between the somatosensory and auditory N2 seems to be difficult. 

Thank you for this helpful comment, and we agree with this. Taking the overlapping of adjacent ERP components into consideration, N2 should be separately analyzed and interpreted. We revised many parts of manuscript.

Methods (from page 10, line 246 to page 11, line 249):

“Judging from grand-averaged ERP waveforms (Fig 1), the amplitude of N2 would be affected by overlapping of adjacent N1. Therefore, N2 components did not directly compare with Modality.”

Results (from page 12, line 293 to page 13, line 309):

“Judging from grand-averaged ERP waveforms (Fig 1), the amplitude of N2 would be affected by overlapping of adjacent N1. Therefore, N2 components did not directly compare with Modality.”

Discussion (from page 19, line 460 to page 20, line 473):

“Furthermore, we found that the peak latency during somatosensory paradigms was significantly earlier in No-go-N2 than in Go-N2, but was not during auditory paradigms (Table 3). In a study of monkeys, Gemba and Sasaki [20] observed No-go-related neural activities after an auditory stimulus in the rostral part of the dorsal bank of the principal sulcus, as opposed to the caudal part of the same bank after a visual stimulus. The present study did not directly address the differences in generator mechanisms of frontal N2 between somatosensory and auditory paradigms, but our findings suggest that the appearance itself of frontal N2 among children does not depend on sensory modalities, and the strength and speed of neural activities involve modality differences. In addition, the difference in the peak latency of No-go-N2 might be related to the developmental difference between somatosensory and auditory processing of motor inhibition.”

Second, the authors should carefully discuss how the physical difference of the Go stimulus and No-go stimulus can affect the ERP and behavioral results. For example, readers would simply wonder how physical differences between the Go stimulus and No-go stimulus can be equalize between the auditory (i.e. , tones of 1000 and 1500 Hz) and somatosensory modalities (i.e., stimulations to second and fifth digits in the left hand) and how such physical differences can affect ERP and behavioral results. This point should be carefully discussed and properly operationalized in the Introduction.

We revised some parts in the Introduction section, as suggested (from page 6, line 147 to page 7, line 153):

“We also designed a Go (target) and No-go (non-target) stimulus with the same probability to avoid the effects of stimulus probability and minimize differences in response conflict between event types [4, 11-13, 33]. These data would advance understanding of the neural development of motor execution and inhibition processing, which is dependent on or independent of the stimulus modality.”

Methods section (from page 8, lines 188 to page 9, line 200):

“In our previous study using the somatosensory Go/No-go paradigms, we set two conditions [4]. In one condition, the Go stimulus was delivered to the second digit of the left hand, and the No-go stimulus to the fifth digit of the left hand. In the other condition, the Go and No-go stimuli were reversed in the left hand, i.e., Go and No-go stimuli were delivered to the fifth and second digits, respectively. As the results, no significant differences between conditions were observed in behavioral data including RT and error rates, nor in the peak amplitudes or latencies of somatosensory ERP components. Therefore, in the present study, we considered that the effects of physical differences between Go and No-go stimuli (i.e., stimulations to second and fifth digits in the left hand; pure tones of 1,500 and 1,000 Hz) on behavioral data and ERP components would be negligible.”

Third, I would not prefer the use of “peak” measures, since it is well known that peak measures are highly vulnerable to high-frequency noises. Although “peak latency” measures are difficult to be replaced with other alternative measures, at least, the authors may use “mean amplitude” measures rather than “peak amplitude”. Anyway, if the authors use “peak” measures, then they should be careful about adverse influences due to high-frequency noises. For example, it was reported that the peak latency of somatosensory P3 was shorter than that of auditory P3 (lines 316-320). This is not consistent with the waveforms in Figure 1. I would suspect that this might be adverse influences of the use of peak latency measure.

Thank you for this helpful comment. We added this possibility as limitation of the present study. In addition, Figure 1 shows simply grand-averaged ERP waveforms across all participants. Therefore, for example, if there are subjects with a large and shorter P3 amplitude, the ERP waveforms could be distorted. This may be related to the difference in appearance between the actual value (i.e., tables) and waveforms (i.e., figures). We also added another possibility as limitation (page 24, lines 567-575):

“The present study used peak measurements for each ERP component. However, since peak measurements may involve high-frequency noises (ex., double peaks of P3), the definition to determine the peak amplitude and latency would be important in this study. In addition, Figure 1 shows simply grand-averaged ERP waveforms across all participants. Therefore, for example, if there are subjects with a large and shorter P3 amplitude, the ERP waveforms could be distorted. This may be related to the difference in appearance between the actual value (i.e., tables) and waveforms (i.e., figures).”

Minor:

Results section (somatosensory N1): As can be seen in Figure 1, somatosensory N1 has the maximum amplitude at the right central electrode (i.e., C4). This seems natural, given that the somatosensory stimuli were presented to the participants’ fingers of the left hand. However, the size and timing of somatosensory N1 were quantified for the Fz electrode. This seems unreasonable.

We　agree with this comment. We revised Methods and Results sections. Methods section (page 10, lines 240-244):

“The amplitudes and latencies of N1s and N1a components at C4 and Fz, respectively, were separately analyzed by one-way repeated measures ANOVA using the within-subject factor of Trial (Go vs. No-go), because N1s and N1a components were the largest at C4 and Fz among electrodes, respectively.”

Results section (from page 12, line 291 to page 13, line 299):

“ANOVAs for the peak amplitude of N1s showed a significant main effect of Trial (F (1, 15) = 6.366, p = 0.023, η2 = 0.298), suggesting larger amplitude of No-go-N1s than that of Go-N1s. ANOVAs for the peak latency of N1s showed no significant main effect of Trial (F (1, 15) = 0.061, p = 0.809, η2 = 0.004). ANOVAs for the peak amplitude and latency of N1a showed no significant main effect of Trial, suggesting no differences between Go and No-go trials (N1a amplitude: F (1, 14) = 1.524, p = 0.237, η2 = 0.098; N1a latency: F (1, 14) = 0.042, p = 0.840, η2 = 0.098) (Table 2).”

Results section (amplitude of P3): It was reported that the peak amplitude of No-go P3 was larger than that of Go P3 during the somatosensory paradigm (lines 312-314). This is not consistent with the waveforms in Figure 1 as well as the values listed in Table 3. Is it correct?

Thank you for your note. It was typo. The amplitude of Go-P3 was larger than that of No-go-P3 during the somatosensory paradigm. We corrected this part (page 14, lines 334-336):

“This two-way interaction indicated that the peak amplitude of Go-P3 was larger than that of No-go-P3 during the somatosensory paradigms”

Results section (scalp distributions): When examining between-condition differences of scalp topographies (e.g., lines 305-307 and 345-348), it is recommended to normalize the amplitude values, for example, by using the method of McCarthy & Wood (1985, Electroencephalography and Clinical Neurophysiology).

We added the normalized amplitude data to clarify the differences in the scalp topography, as suggested (page 14, lines 323-328):

“In addition, we examined the normalized amplitude values to clarify the differences in the scalp topography [37]. ANOVAs revealed a significant main effect of Electrode (F (2, 34) = 178.071, p < 0.001, η2 = 0.913), and Trial-Electrode interaction (F (2, 34) = 6.320, p = 0.005, η2 = 0.271). This interaction indicates the different distribution between Go and No-go trials (Fig 2).”

We also revised Figure 2.

Results section (general): “p > .05” (where statistical differences were not significant) is not informative for readers. It would be better to simply show p-values. Also, effect sizes should be shown for all statistical tests.

Reviewer #1 also pointed out the same issue. We revised the Results section (from page 12, line 274 to page 16, line 370).

Figure 1: ERPs are shown in a “negative-up” manner. This should be explicitly described in the figure or the figure caption.

Revised (page 33, lines 859-860):

“In the waveforms, the top is shown as negative, and the bottom is shown as positive.”

We also revised Figure 1.

Figure 1: Please show the baseline (i.e., x-axis) with time scales in each of ERP waveforms. It should be helpful for readers to visually evaluate the size and timing of each ERP component.

We revised Figure 1.

Limitation of the present study: I think that the lack of visual Go/No-go paradigm would not be a limitation of the present study.

Thank you for this suggestion. We deleted this part.

---

## [Decision Letter · Decision Letter 1]

25 Oct 2021

Modality differences in ERP components between somatosensory and auditory Go/No-go paradigms in prepubescent children

PONE-D-21-09252R1

Dear Dr. Nakata,

We’re pleased to inform you that your manuscript has been judged scientifically suitable for publication and will be formally accepted for publication once it meets all outstanding technical requirements.

Kind regards,

Vilfredo De Pascalis

Academic Editor

PLOS ONE

Additional Editor Comments (optional):

Both reviewers are satisfied with the revised manuscript. The authors addressed all the raised issues. Thus, I am glad to accept the current version of the manuscript for publication.

Reviewers' comments:

Reviewer's Responses to Questions

**Comments to the Author**

1. If the authors have adequately addressed your comments raised in a previous round of review and you feel that this manuscript is now acceptable for publication, you may indicate that here to bypass the “Comments to the Author” section, enter your conflict of interest statement in the “Confidential to Editor” section, and submit your "Accept" recommendation.

Reviewer #1: All comments have been addressed

Reviewer #2: All comments have been addressed

2. Is the manuscript technically sound, and do the data support the conclusions?

Reviewer #1: Yes

Reviewer #2: Yes

3. Has the statistical analysis been performed appropriately and rigorously? 

Reviewer #1: Yes

Reviewer #2: Yes

4. Have the authors made all data underlying the findings in their manuscript fully available?

Reviewer #1: Yes

Reviewer #2: Yes

5. Is the manuscript presented in an intelligible fashion and written in standard English?

Reviewer #1: Yes

Reviewer #2: Yes

6. Review Comments to the Author

Reviewer #1: All the points that I rised have been addressed by the Authors. It is my opinion that now the manuscript is nice!

Reviewer #2: (No Response)

7. PLOS authors have the option to publish the peer review history of their article (what does this mean?). If published, this will include your full peer review and any attached files.

Reviewer #1: **Yes: **Carlo Lai

Reviewer #2: **Yes: **Motohiro Kimura

---

## [Editor Report · Acceptance letter]

28 Oct 2021

PONE-D-21-09252R1 

Modality differences in ERP components between somatosensory and auditory Go/No-go paradigms in prepubescent children 

Dear Dr. Nakata:

I'm pleased to inform you that your manuscript has been deemed suitable for publication in PLOS ONE. Congratulations! Your manuscript is now with our production department. 

Kind regards, 

on behalf of

Prof. Vilfredo De Pascalis 

Academic Editor

PLOS ONE